# The geography of measles vaccination in the African Great Lakes region

Saki Takahashi[1], C. Jessica E. Metcalf[1,2], Matthew J. Ferrari[3], Andrew J. Tatem[4,5] & Justin Lessler[6]

Expanded access to measles vaccination was among the most successful public health interventions of recent decades. All WHO regions currently target measles elimination by 2020, yet continued measles circulation makes that goal seem elusive. Using Demographic and Health Surveys with generalized additive models, we quantify spatial patterns of measles vaccination in ten contiguous countries in the African Great Lakes region between 2009–2014. Seven countries have 'coldspots' where vaccine coverage is below the WHO target of 80%. Over 14 million children under 5 years of age live in coldspots across the region, and a total of 8–12 million children are unvaccinated. Spatial patterns of vaccination do not map directly onto sub-national administrative units and transnational coldspots exist. Clustering of low vaccination areas may allow for pockets of susceptibility that sustain circulation despite high overall coverage. Targeting at-risk areas and transnational coordination are likely required to eliminate measles in the region.

[1] Department of Ecology and Evolutionary Biology, Princeton University, Princeton, New Jersey 08544, USA. [2] Woodrow Wilson School of Public and International Affairs, Princeton University, Princeton, New Jersey 08544, USA. [3] Center for Infectious Disease Dynamics, The Pennsylvania State University, State College, Pennsylvania 16802, USA. [4] WorldPop, Department of Geography and Environment, University of Southampton, Southampton SO17 1BJ, UK. [5] Flowminder Foundation, Stockholm SE-11355, Sweden. [6] Department of Epidemiology, Johns Hopkins Bloomberg School of Public Health, Baltimore, Maryland 21205, USA. Correspondence and requests for materials should be addressed to J.L. (email: justin@jhu.edu).

A potentially high case-fatality rate, combined with the existence of an inexpensive and safe vaccine that provides lifelong immunity, makes measles control one of the most cost-effective public health interventions in existence[1,2]. Due to substantial gains in measles vaccination coverage over recent decades, incidence has fallen worldwide from an estimated 146 cases per million population and 562,400 deaths in 2,000, to 40 cases per million population and 114,900 deaths in 2014 (ref. 3). However, measles continues to circulate in many countries and remains one of the leading killers of children globally[4].

The target vaccination coverage that must be reached to achieve measles elimination is a function of how efficiently the virus is spread, which is in turn a result of the biology of measles and the contact patterns of infected individuals. The efficiency of viral spread is captured by the basic reproductive number $R_0$, defined as the number of secondary cases an infected individual would cause in a fully susceptible population (estimated to be between 10 and 20 for measles[5,6]). In the simplest analysis, measles requires at least $1/R_0$ of a population to be susceptible to measles in order for the virus to persist. Hence for measles, between 90 and 95% of the population must be immune to interrupt measles transmission. However, this analysis is based on the assumption that unvaccinated individuals are evenly distributed throughout the population, which is unlikely to be true in the real world.

Measles is a virus transmitted mainly through direct contact (coughing and sneezing) and also by small-particle aerosols in an airspace where an infected person has coughed or sneezed in the previous few hours[7]. Infected individuals must enter into one of these forms of 'contact' with susceptible individuals during the ∼2 weeks that they are infectious in order for a chain of transmission to persist[8]. Patches of unvaccinated individuals living in close proximity are therefore more likely to sustain a measles epidemic compared to the same number of unvaccinated people evenly distributed throughout a country. Even when the size of these unvaccinated clusters is below the critical community size required to maintain measles transmission (estimated to be around 300,000 individuals for measles in pre-vaccination England and Wales[9], though differences in birth rate and population density may lead to differences in the African context[10]), they remain a concern: even transient outbreaks can cause significant morbidity and mortality, and seeding of new outbreaks by movement between clusters can potentially maintain regional transmission.

The impact of spatially heterogeneous vaccination has been increasingly recognized in making policy decisions, resulting in a shift in focus from simply setting country-level targets for coverage, to ensuring uniformly high vaccination levels across countries (for example, the strategy of RED (Reaching Every District)[11]). Although a considerable improvement over a country-level focus, a district-level focus may still miss important aspects of geographical heterogeneity. By taking averages across administratively-defined areas (that is, provinces or districts), we may miss zones of vulnerability that are small or do not reflect national or sub-national administrative boundaries.

Throughout sub-Saharan Africa, where the majority of the world's remaining measles burden is found[12], measles vaccination is predominantly delivered through two activities: routine immunization (that is, at local health centres) that target children around 9 months of age for their first dose of measles-containing vaccine (MCV-1)[13], and supplemental immunization activities (SIAs), which are large campaigns periodically conducted that target a broader age range in an attempt to provide a second vaccine dose to those vaccinated in routine programs and to provide a first dose to those who were not. This two-pronged approach was successful in the Americas, which achieved endemic measles elimination in 2002 (ref. 14; although cases have continued to be imported in the region; for example, after elimination was declared in Brazil in 2000, there was an average of 50 reported cases per year between 2001–2014 (ref. 15)).

The combination of increased routine vaccination coverage and periodic SIAs reduced yearly measles incidence in Africa by 93% between 2001–2008 from 492,000 to 37,000 reported cases[16,17]. However, since mid-2009 there has been a resurgence, with ∼200,000 measles cases reported in 28 countries in sub-Saharan Africa between 2009–2010 (refs 18,19). Such outbreaks have been attributed to weak routine vaccination systems and delayed or low-quality SIA campaigns[20], along with 'honeymoon period' effects as a possible synergistic mechanism[21]. A lack of transnational coordination in the timing of SIAs may also contribute (for example, Mozambique conducted SIAs in 2008, 2011 and 2013, while neighbouring Zimbabwe did so in 2009, 2010 and 2012), potentially allowing the virus to persist in spatial clusters of unvaccinated children that cross international boundaries.

Demographic and Health Surveys (DHS) provide cross-sectional data on the spatial distribution of vaccination in children under 5 years of age across multiple countries, collected using a standardized framework[22]. Combining this data with information on the age and geographic distribution of the local population, we here map vaccine-derived immunity against measles in ten East African countries in the African Great Lakes region that use SIA campaigns to boost population-level immunity: the Democratic Republic of Congo (DRC), Uganda, Kenya, Rwanda, Burundi, Tanzania, Zambia, Malawi, Mozambique and Zimbabwe. Using these maps we identify coldspots of measles vaccination that cross administrative boundaries, foci for elimination efforts and locations where elimination efforts may be failing. In doing so, we aim to inform the spatial scale at which vaccination policy could be most effectively implemented in the region (nationally, sub-nationally or along which administrative borders), and highlight the complexities and challenges associated with current approaches.

## Results

**Vaccination coverage.** We estimated measles vaccination coverage at each month of age across the region using generalized additive models (GAMs)[23] (Methods section) with DHS surveys conducted between 2009–2014. Estimated vaccination coverage is the result of both routine immunization programs and national SIA campaigns for which children are eligible, given their age. Taking vaccination coverage at 24 months of age as a representative scenario, results indicate large contiguous areas of low vaccination coverage across the region (Fig. 1a). In particular, in DRC, low vaccination areas are found in the northwest (former Equateur province), central (former Kasaï-Occidental and Kasaï-Oriental provinces) and southeast (former Katanga province). Other countries show greater overall variability: within-country heterogeneity in coverage is particularly pronounced in Tanzania and Mozambique, while Rwanda, Burundi and Malawi have relatively homogeneous- and high-vaccine coverage. In Kenya, vaccination coverage decreases with greater distance from the capital, Nairobi; the opposite qualitative pattern is found in Zimbabwe and in Uganda, with decreased coverage with proximity to the capital cities.

In our main results, we did not attempt to map the impact of sub-national SIA campaigns because variability in their age-eligibility (Supplementary Fig. 3) complicates the interpretation of within-country spatial patterns. However, vaccination coverage maps including the impact of sub-national campaigns are not qualitatively different (Supplementary Fig. 13,

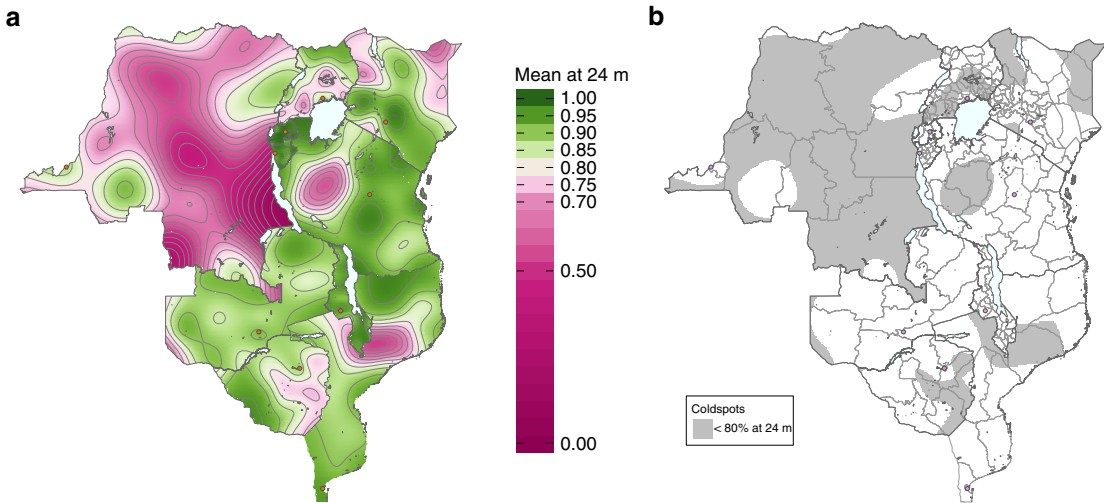

**Figure 1 | Vaccination coverage and coldspots at 24 months of age.** (**a**) Estimated mean proportion of children 24 months of age who have either received routine measles vaccination or were vaccinated during a national measles SIA campaign. Contour lines are marked at every 0.05 level. Capital cities are shown as orange circles. (**b**) Estimated coldspots (defined as below 80% estimated mean measles vaccination coverage) of routine and national SIA measles vaccination for children 24 months of age. Capital cities are shown as pink circles. The first sub-national political boundaries are shown in light grey.

| Table 1 \| DHS survey data and SIA campaigns. | | | | | | | |
|---|---|---|---|---|---|---|---|
| Country | DHS survey start date | DHS survey end date | Number of children in survey, 6–60 months | Number of GPS clusters in survey | Number of national SIA campaigns, with eligibles in DHS | Number of sub-national SIA campaigns, with eligibles in DHS | Proportion of vaccinations ($v = 1$) based on vaccination card |
| Burundi | 08/2010 | 01/2011 | 6,661 | 376 | 4 | 1 | 0.4519 |
| DRC | 08/2013 | 02/2014 | 14,321 | 492 | 0 | 20 | 0.1406 |
| Kenya | 05/2014 | 10/2014 | 18,311 | 1,583 | 1 | 0 | 0.6198 |
| Malawi | 06/2010 | 09/2010 | 16,379 | 827 | 2 | 0 | 0.6281 |
| Mozambique | 05/2011 | 12/2011 | 9,369 | 609 | 2 | 0 | 0.7042 |
| Rwanda | 09/2010 | 04/2011 | 7,883 | 492 | 2 | 0 | 0.7032 |
| Tanzania | 12/2009 | 05/2010 | 6,592 | 458 | 0 | 1* | 0.7440 |
| Uganda | 06/2011 | 12/2011 | 6,580 | 400 | 1 | 0 | 0.4636 |
| Zambia | 08/2013 | 04/2014 | 11,659 | 719 | 2 | 0 | 0.6358 |
| Zimbabwe | 09/2010 | 03/2011 | 4,494 | 393 | 2 | 0 | 0.4737 |

DHS, Demographic and Health Surveys; DRC, Democratic Republic of Congo; SIA, supplemental immunization activity.
Description of DHS data and SIA campaign information included in the analysis, by country.
*Tanzania had an SIA campaign in 08/2008–09/2008 that targeted all mainland parts of the country, so this campaign is considered to be sub-national.

Supplementary Tables 4 and 8). Most countries only had one or two national SIAs over the period of interest, with the exceptions of Burundi with four national SIAs and one sub-national SIA, DRC with no national SIAs and 20 sub-national SIAs, and Tanzania with no national SIAs and one sub-national SIA (targeting all mainland parts of the country in a multi-day campaign)[24] (Table 1). Model residual variograms do not suggest existence of unmodelled spatial autocorrelation (Supplementary Fig. 5).

**Coldspots of vaccination**. To define key areas of low coverage, we first defined a coldspot of vaccination as a grid cell that has below 80% estimated mean measles vaccination coverage at a given age (Methods section). We then mapped the coldspots of measles vaccination at 24 months of age, indicating areas from Fig. 1a where mean coverage is estimated to be below 80%, in grey (Fig. 1b). Across the region, coldspots span multiple sub-national administrative units (the largest sub-national political boundaries

(Adm 1) shown in light grey). There were no or very few coldspots in Rwanda, Burundi, Malawi and Zambia. Tanzania and Kenya have large areas covered by coldspots; however, only low percentages (<9%) of children reside within them, as coldspots exist in low population density locations (Table 2). Conversely, high percentages of children in DRC and Uganda live in a coldspot: 70 and 46%, respectively. In DRC we identified large areas covered by coldspots. Summing over all ages under 5 years, we found that 7,844,517 (95% confidence interval (CI): 5,688,449–9,221,993) children in DRC and 3,033,139 (95% CI: 401,684–4,915,302) children in Uganda reside within a coldspot defined at 24 months of age. We also mapped the confidence in our classification of each grid cell as a coldspot, incorporating the s.e. (Supplementary Fig. 9, from Supplementary Figs 6–8).

**Size of unvaccinated population**. Estimates of vaccination coverage were combined with data on population size to map the numbers of children between 6–24 months of age who were not

| Table 2 | Children residing in coldspots and unvaccinated children. | | | |
|---|---|---|---|
| Country | Percentage of children under 60 months of age, who reside in a coldspot defined at 24 months (95% CI) | Number of children under 60 months of age, who reside in a coldspot defined at 24 months (95% CI) | Number of unvaccinated children, 6–24 months of age (95% CI) | Number of unvaccinated children, 6–60 months of age (95% CI) |
| Burundi | 0.00 (0.00–0.27) | 0 (0–4,488) | 156,115 (136,781–179,535) | 191,809 (156,170–244,804) |
| DRC | 70.17 (50.88–82.49) | 7,844,517 (5,688,449–9,221,993) | 1,753,999 (1,549,022–1,966,182) | 3,560,359 (2,906,664–4,305,564) |
| Kenya | 8.65 (4.48–11.81) | 584,202 (302,478–797,303) | 751,455 (693,565–813,806) | 1,193,622 (1,047,857–1,361,819) |
| Malawi | 0.00 (0.00–0.00) | 0 (0–0) | 253,333 (225,987–282,985) | 341,742 (289,379–405,658) |
| Mozambique | 29.02 (18.32–41.49) | 1,229,388 (776,359–1,757,885) | 518,287 (455,962–586,121) | 852,445 (705,614–1,029,626) |
| Rwanda | 0.00 (0.00–0.00) | 0 (0–0) | 166,891 (150,075–188,445) | 200,079 (168,233–248,821) |
| Tanzania | 8.63 (4.44–21.81) | 693,420 (356,989–1,753,481) | 970,427 (854,019–1,110,930) | 1,432,727 (1,134,152–1,857,962) |
| Uganda | 46.12 (6.11–74.75) | 3,033,139 (401,684–4,915,302) | 905,064 (795,970–1,024,601) | 1,495,953 (1,227,860–1,821,788) |
| Zambia | 0.80 (0.08–7.86) | 19,222 (1,883–189,951) | 287,954 (259,420–319,670) | 407,544 (345,988–483,536) |
| Zimbabwe | 31.88 (0.00–63.47) | 621,486 (0–1,237,487) | 255,745 (220,942–294,788) | 457,208 (361,951–576,215) |
| Total | 29.62 (15.90–41.98) | 14,025,374 (7,527,842–19,877,890) | 6,019,270 (5,341,773–6,767,063) | 10,133,488 (8,343,868–12,335,793) |

CI, confidence interval; DRC, Democratic Republic of Congo; GAM, generalized additive model; SIA, supplemental immunization activity.
Estimated percentage and number of children under 60 months of age who reside in measles vaccination coldspots (for routine and national SIAs) defined at 24 months of age, and estimated number of children 6–24 months of age and 6–60 months of age who have neither received routine measles vaccination nor were vaccinated during a national measles SIA campaign, by country and total region. 95% CI from the standard errors of GAM predictions.

vaccinated against measles in routine activities or national SIAs (Fig. 2). Numbers of unvaccinated children in these countries range from 156,115 (95% CI: 136,781–179,535) in Burundi to 1,753,999 (95% CI: 1,549,022–1,966,182) in DRC, with an estimated total of 6,019,270 (95% CI: 5,341,773–6,767,063) children between 6–24 months of age and 10,133,488 (95% CI: 8,343,868–12,335,793) children between 6–60 months of age across the entire region (Table 2). Although Rwanda, Burundi and Malawi attain high levels of vaccination coverage, these countries are densely populated and thus still have high numbers of unvaccinated children. The map of unvaccinated children in Fig. 2 also shows that despite central DRC having the lowest vaccination coverage and largest coldspots by area as shown in Fig. 1, large numbers of unvaccinated individuals cluster elsewhere in the region. Notably, a substantial transnational cluster of unvaccinated children exists in the densely populated region surrounding Lake Victoria (including areas of relatively high vaccination coverage). Estimated numbers of unvaccinated children and proportions vaccinated by age group is provided in Supplementary Fig. 11, Supplementary Tables 6 and 7.

**Coldspots of vaccination and population density.** Because vaccination coverage varies by age (Supplementary Movie 1), the locations of areas considered to be coldspots vary by age as well (Supplementary Movie 2). We identified 'long-term' coldspots, or grid cells where vaccination coverage is below 80% over a large proportion of monthly age cohorts between 12–60 months (Fig. 3a), shown in dark red. The spatial patterns are similar to those in Fig. 1a: there are large areas of DRC that are long-term coldspots. However, Fig. 3a captures only the estimated levels of vaccination coverage and does not account for population density, which is another factor that influences measles risk. Therefore, to obtain a complete picture of what Fig. 3a indicates in the context of population size, we further partitioned the long-term coldspots into 'low-density' and 'high-density' areas: only coldspots with at least 500 children under 60 months of age per grid cell are shown in Fig. 3b. In Uganda we observed a large overlap between coldspots and high population density areas. The percentage of total grid cells with at least 500 children ranged from 7% in Zambia to 87% in Burundi (Supplementary Table 10). The dark red grid cells in Fig. 3b now represent long-term, high-density coldspots (sensitivity analysis to this cutoff value of 500 is provided in Supplementary Fig. 12). This indicates that many of the long-term coldspots are likely of limited epidemiological importance because few people live there: the

long-term, high-density coldspots should be a priority due to both low vaccination coverage and significant numbers of unvaccinated children.

**Spatial variation within and between countries.** Up to this point, we have focused on identifying epidemiologically relevant areas (that is, areas highlighted by vaccination coverage itself). However, as control strategies are usually designed in the context of political boundaries, we also quantified the relative contributions of individual sub-national political boundary levels (that is, programmatically relevant spatial scales) to the overall variance in measles vaccination coverage within each country (Table 3) (see equation (2) in the Methods section). In seven out of the ten countries, the largest sub-national administrative level (provinces) explained the largest proportion of the overall variance; the largest two sub-national administrative levels together (provinces and districts) accounted for a majority of the variance within each country.

Finally, we quantified the spatial variation between countries by aggregating data from the ten countries and analysing the data as a single region. We employed the same GAM framework (see equation (1) in the Methods section) for this aggregated data with an additional covariate for country, and found transnational heterogeneities in coverage (Supplementary Table 5). Then in a multi-level modelling framework (see equation (3) the Methods section), we estimated that 59% (95% CI: 28–73%) of the variation in the probability of being vaccinated at any given age is explained by the country in which the individual lives and that there is still considerable sub-national variation in coverage.

**Discussion**
Our results indicate significant heterogeneity in measles vaccine coverage within and between the ten countries examined, encompassing an estimated total of 10 million unvaccinated children between 6–60 months of age. Large coldspots where high proportions of children remain unvaccinated against measles through their fifth year of life are found across the region, particularly in countries with low overall coverage (for example, DRC). The over 14 million children who live in these coldspots, particularly those too young to be vaccinated but no longer protected by maternal antibodies, do not benefit from the herd protection provided by high population vaccination coverage. A focus on coverage and coldspots alone does not give a full picture of the vaccination landscape, as the numbers of individuals at these locations must also be evaluated. We show

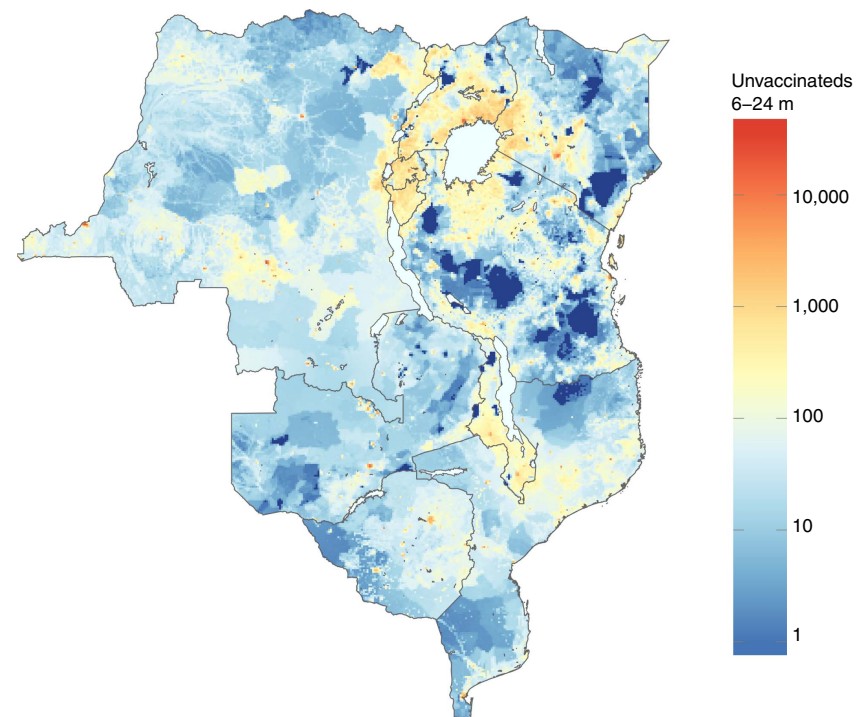

**Figure 2 | Unvaccinated children 6–24 months of age.** Estimated number of children 6–24 months of age per 10 km by 10 km grid cell who have neither received routine measles vaccination nor were vaccinated during a national measles SIA campaign. Dark blue grid cells have estimated zero population density.

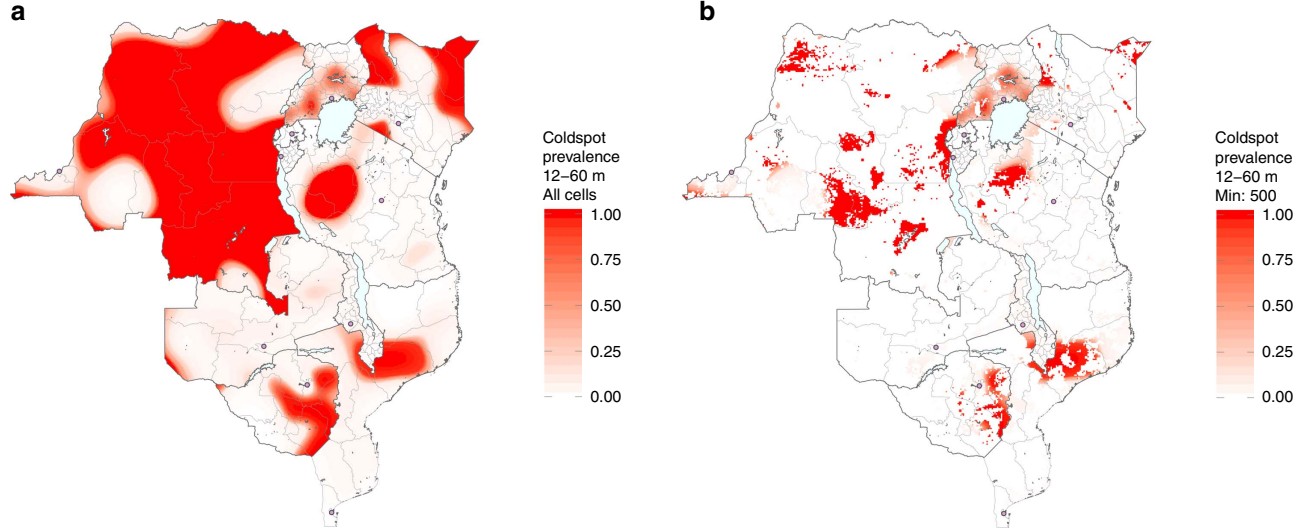

**Figure 3 | Vaccination coldspots and population density.** Estimated proportion of monthly age cohorts that each 10 km by 10 km grid cell exists as a coldspot of routine and national SIA measles vaccination for children between 12–60 months of age (total of 49 monthly age cohorts), showing (**a**) all grid cells (long-term coldspots) and (**b**) only grid cells with at least 500 children under 60 months of age (long-term, high-density coldspots). Capital cities are shown as pink circles. The first sub-national political boundaries are shown in light grey.

that the high population density and birth rates around Lake Victoria and throughout the African Great Lakes region mean that even areas with relatively high vaccination coverage have large numbers of unvaccinated children. Areas where low coverage and high population density combine should be top priorities for stepped-up immunization efforts for measles control, and become particularly important in the context of measles elimination.

The approach taken here uses standard techniques with the goal that it could be easily applied across different settings. Country-level estimates of vaccination (Supplementary Table 12) often average across important sources of heterogeneity, and the methods presented here can be used to identify and visualize sub-national coldspots of vaccination (which likely correspond to geographic clusters of susceptibility) that cross administrative boundaries, which may not be evident in analyses solely based on

**Table 3 | Sub-national variation in vaccination coverage.**

| Country | Adm 1 ICC (95% CI) | Adm 2 ICC (95% CI) | Adm 3 ICC (95% CI) | Residual ICC (95% CI) |
|---|---|---|---|---|
| Burundi | **0.5798** (0.3169–v0.7058) | 0.1668 (0.0891–0.3015) | <0.0001 (0.0000–0.0723) | 0.2535 (0.1615–0.4049) |
| DRC | 0.2404 (0.0000–0.4656) | **0.3582** (0.1685–0.5681) | 0.2856 (0.1969–0.4422) | 0.1158 (0.0867–0.1779) |
| Kenya | **0.8271** (0.7478–0.8763) | 0.1044 (0.0725–0.1537) | 0.0354 (0.0250–0.0530) | 0.0331 (0.0241–0.0483) |
| Malawi | **0.5015** (0.3187–0.6203) | 0.2759 (0.2013–0.3888) | 0.0064 (0.0000–0.0315) | 0.2163 (0.1635–0.2998) |
| Mozambique | **0.5506** (0.2213–0.7106) | 0.3736 (0.2391–0.6481) | 0.0443 (0.0290–0.0811) | 0.0315 (0.0209–0.0557) |
| Rwanda | 0.3186 (0.0000–0.5849) | **0.4210** (0.2100–0.6896) | 0.1468 (0.0842–0.2848) | 0.1136 (0.0693–0.2190) |
| Tanzania | **0.7474** (0.6063–0.8253) | 0.1364 (0.0895–0.2197) | 0.0556 (0.0388–0.0881) | 0.0606 (0.0434–0.0948) |
| Uganda | **0.7765** (0.6782–0.8347) | 0.1018 (0.0648–0.1583) | 0.0404 (0.0281–0.0608) | 0.0813 (0.0613–0.1145) |
| Zambia | 0.3469 (0.0679–0.5338) | **0.4055** (0.2630–0.5982) | NA | 0.2476 (0.1812–0.3769) |
| Zimbabwe | **0.6098** (0.2145–0.7668) | 0.2274 (0.1196–0.4874) | NA | 0.1628 (0.1008–0.3255) |

Adm, administrative (level); CI, confidence interval; DRC, Democratic Republic of Congo; ICC, intraclass correlation coefficient; NA, not available.
ICC of the top three sub-national political boundary levels (Adm), by country. Adm 1 refers to provinces, Adm 2 refers to districts, and Adm 3 refers to municipalities. Adm level with largest contribution to overall variance shown in bold. 95% CI from quantiles of parametric bootstrap distributions. NA if the country has fewer than three Adm levels.

these divisions. Characterizing such heterogeneities is especially important as we strive for measles elimination. Translating these vaccination coldspots into specific interventions will require a formal evaluation of the operational and logistical challenges of spatially targeted efforts, but coordinated cross-political boundary efforts are likely to be a critical element of success.

Our analysis suggests that targeting of efforts at the largest sub-national administrative unit (usually provinces) would account for the majority of sub-national variation in vaccination coverage (Table 3). This indicates that, in terms of strengthening measles vaccination programs, targeting broad spatial groupings may be effective for the countries here. However, while these large units explained the largest proportion of the overall variability in vaccination coverage, the dynamics of measles incidence will be shaped by local heterogeneities that do not necessarily reflect these boundaries[25] and thus cross-province coordination or more focused targeting may be needed depending on the situation.

Spatial clustering of unvaccinated individuals may lead to pockets of measles susceptibility that will sustain circulation, even in an otherwise successful measles elimination programme with high overall vaccination coverage[26]. Age cohorts missed by routine vaccination in each year will allow continued circulation of the virus, unless they are removed from the pool of susceptibles by natural infection or a broader age range campaign like an SIA. Furthermore, insufficient vaccination coverage may increase the average age of infection such that children with no immunity (neither from vaccination nor from infection) will become the focus of adolescent and adult measles outbreaks in the future[27]. From a programmatic perspective it will be important to consider the specific effects of varying the lower and upper age targets of potential campaigns, and our analysis suggests that these may be spatially context-specific. Our estimates of spatial heterogeneity might thus be leveraged to aid in the design of sub-national vaccination campaigns.

The heterogeneities in proportions unvaccinated by age revealed by our analysis illustrate that coverage patterns are not static and vary across age cohorts (Supplementary Movies 1 and 2). Though the worst-performing areas appear to consistently have poor coverage throughout all birth cohorts, transient coldspots that are only present for a few years may still create important pockets of susceptibility that can later cause problems for measles control or be a sign of an emergent problem. Ideally, these age profiles of susceptibility could be used to guide control measures. For instance, coldspots of vaccination among older children might suggest the importance of continued SIAs as a key component of efforts to mitigate problems with geographic clustering of measles susceptibility, while coldspots among younger children might suggest the need to strengthen routine health care

systems since clusters of measles risk will quickly re-form among young children as a function of the local birth rate.

In addition to proportions unvaccinated by age (which directly translate to coldspots), population density is also an important factor that influences measles risk, since more densely populated areas are more likely to sustain epidemics than less densely populated areas[9]. Areas with both low vaccination coverage across ages and a high density of children are likely to have a large pool of susceptible individuals, and are thus at greatest risk for an outbreak. Furthermore, the absolute numbers of unvaccinated children (as illustrated in Fig. 2) may have a policy relevance independent of risk, as they highlight areas where targeting vaccination efforts might be most cost-effective[28].

There is some empirical evidence that the patterns uncovered here are relevant to mapping measles risk and informing control. Recent measles outbreaks in DRC followed spatial patterns consistent with our identified coldspots and high-density, low vaccination coverage areas[29]. However, translating maps of unvaccinated children into maps of susceptible children is complicated by the acquisition of immunity via natural infection[30]. For example, Zambia had a major measles outbreak within the 5 years before its DHS survey[31] while other countries did not, with important implications for susceptibility that would be complemented by study of age-stratified measles incidence and/or serological data from the region. An additional limitation here is that we do not include aspects such as the effect of school-term on seasonality[32] or the impact of schools as hotspots of transmission[33], both of which are sources of non-geographic heterogeneities of mixing that are important to understanding the dynamics of observed measles incidence. Furthermore, DHS surveys are not all conducted in the same year, which makes interpretation of between-country comparisons difficult.

We also currently do not distinguish between immunity acquired through routine vaccination and through national SIAs. Disentangling the effects of routine programs, national SIAs and sub-national SIAs on overall vaccination coverage may provide important programmatic information and is a key direction for future work. Methodological refinements in a model-based geostatistical framework (for example, the Malaria Atlas Project[34] and the WorldPop project[35]) and the inclusion of spatial metrics such as urbanicity, remoteness or accessibility[36] will strengthen model fit (Supplementary Tables 1 and 2) and capture smaller-scale spatial variability that is missed by this current approach. However, larger-scale patterns are consistent (which is our main focus here), given that the addition of the DHS urban-rural specification to our model yields congruent maps (Supplementary Fig. 15). In addition, it will be important to identify and quantify the effects of logistical issues such as vaccine stock outs[37] or

socioeconomic factors[38] that may also affect the performance of immunization programs. The increasing abundance of methods development and applications of spatial interpolation for DHS and similar data[39–44] broadly highlight the utility of maps as a tool for public health planning and advocacy. For example, the DHS Program has recently created a Spatial Data Repository[45,46] with interpolated surfaces of health indicators, including measles vaccination coverage among children 12–23 months of age, in a Bayesian model-based geostatistical framework with spatial covariates. While we use the same data source in this analysis to model a similar outcome, our approach here differs in that we fit an age-varying probability of vaccination.

Spatial heterogeneity in measles vaccination coverage raises a further key public health policy issue: many of the countries investigated here will soon become eligible for Global Alliance for Vaccines and Immunization funding to support the introduction of rubella-containing vaccine[47]. Rubella is a mild infection unless contracted by pregnant women during their first trimester, which can lead to the birth of a child with congenital rubella syndrome (CRS). As previous work has shown, inequities in vaccination can lead to an increased risk of rubella infection in pregnant women and thus a higher burden of CRS[48]. Since rubella-containing vaccine is usually given in conjunction with MCV, the spatial heterogeneity in measles vaccination coverage documented here could affect the dynamics of rubella in ways that might increase the burden of CRS in this region[49,50].

The considerable spatial heterogeneities and geographic clustering of low vaccine coverage areas found in our analysis suggest that countries with high levels of national coverage may still be at considerable risk for measles outbreaks. Areas where there is a confluence of high population density and low vaccination coverage (as illustrated in Fig. 3b) pose the greatest risk, and if linked, may have the potential to sustain measles transmission regionally despite robust vaccination campaigns nearly everywhere else. If countries can identify and eliminate these high-risk vaccination coldspots, they will reduce their risk of measles outbreaks and accelerate progress towards the goal of measles elimination.

## Methods

**Data.** Country-level data on measles vaccination status was extracted from the most recent geo-located DHS survey made publicly available by ICF International[51–60]. Survey periods ranged from December 2009 to October 2014 (Table 1). A national DHS survey has one record for each interviewed woman's child 5 years of age and younger at the time of the survey (Supplementary Fig. 2), and is linked to a database of GPS coordinates (longitude and latitude) of respondents' home locations. GPS coordinates are aggregated into clusters containing ∼20 households (Supplementary Fig. 1), and randomly displaced up to 2 km in urban areas and up to 5 km in rural areas (an additional 1% of rural clusters are displaced up to 10 km) to protect respondent confidentiality[61]. For the purposes of this analysis we assume cluster locations are exact. In sensitivity analysis we find the effects of this random displacement to be negligibly small (Supplementary Fig. 16).

For each child, we obtained the age at the time of survey, whether the child had ever received a measles vaccine (based on vaccination card or report of parent/guardian), and the GPS location. Ages were rounded up into 1-month classes due to uncertainty in the data, and children under 6 months of age at the time of survey were considered not to be 'at risk' for successful vaccination and excluded from the analysis (Supplementary Table 9): while routine vaccination with MCV-1 is recommended at 9 months of age, SIA campaigns often set their lower age target at 6 months.

Issues linked to parental recall of vaccination status make this source of information potentially less reliable than card-based validation. Parental recall does not provide the exact date at which a child was vaccinated, and cannot be used to distinguish between vaccination obtained via either the routine programme or SIA campaigns. However, because vaccination cards are often unavailable in the database (for example, they may be lost; Table 1), using parental recall along with vaccination cards allows for greater spatial and temporal scope. Parental recall has been shown to provide a relatively robust indicator of vaccination status in other analyses[62] and was appropriate to our needs, as vaccine-derived immunity (whether from the routine programme or SIAs) and its spatial heterogeneity was our main focus here. We do not use the information on a child's vaccine date in this analysis.

Information on the timing and spatial scale of SIAs was obtained primarily from the WHO[63], with supplementary information on specific campaigns provided by field reports (for example, SIAs conducted in DRC in 2009 and 2011)[64]. SIA campaigns can either be conducted on a national scale or target one or more sub-national regions. In this analysis, we included all SIAs that were completed within the 5 years before a country's DHS survey (Table 1, Supplementary Tables 4 and 11), and removed the outbreak response immunization campaigns which are conducted at smaller spatial scales[65]. Spatially structured, population demographic data from 2010 by 5-year age groupings was obtained from the WorldPop project[66,67] (Supplementary Fig. 10) and monthly age groupings were obtained assuming a uniform distribution. National and sub-national administrative political boundary shapefiles were obtained from DIVA-GIS[68]. All analysis was conducted using the R statistical software, version 3.2.3 (http://cran.r-project.org).

**Definition of coldspots.** One of the milestones for 2015 established in 2010 by the World Health Assembly was to increase routine coverage with MCV-1 for children aged 1 year to at least 90% nationally and at least 80% in every district by every member state[69]. Hence, in this analysis, we define a coldspot of vaccination to be a spatial unit (that is, grid cell) that has below 80% estimated mean measles vaccination coverage at a given age. The definition of a coldspot is therefore age-specific.

**Estimating vaccination coverage.** We performed logistic regression using GAMs to estimate measles vaccination coverage separately for each country, where the outcome was whether a child $i$ was reported as vaccinated ($v_i = 1$) or not ($v_i = 0$) in the DHS survey. To account for spatial autocorrelation, we included longitude and latitude (long$_i$, lat$_i$) as a smoothed ($s$) interaction term in the GAM[70–72]. We also included monthly survey age (age$_i$) as a smoothed covariate (Supplementary Table 3), which showed a broadly increasing relationship with vaccination coverage (Supplementary Fig. 4). We did not explicitly model national SIAs (Supplementary Table 11), as a child's eligibility for being vaccinated during a national campaign is collinear with survey age. However, if a country had any sub-national SIAs during the time period of interest (here, Burundi, DRC and Tanzania), we included eligibility of individual $i$ for each sub-national campaign $j$ ($c_{ij}$) as a covariate (where eligibility is based on both survey age and geographic location). The GAM used is shown in equation (1)

$$\text{logit}(v_i) = s(\text{long}_i, \text{lat}_i) + s(\text{age}_i) + \sum_j c_{ij} \qquad (1)$$

We do not use the DHS sample weights here, as we do not calculate any aggregate or national measures (robustness to inclusion of sample weights shown in Supplementary Fig. 17). Equation (1) reflects the primary model used in this analysis, and the Akaike information criterion values of sub-models is presented in Supplementary Tables 1 and 2 (also see Supplementary Fig. 14). Vaccination coverage levels were determined by laying a 10 km by 10 km grid across the country and interpolating the expected value for each grid cell, and then combined with age-structured, spatially explicit data on the population size of children. The distribution of the population density of children under 5 years of age by country is provided in Supplementary Table 10.

**Sub-national clustering of susceptibility.** At the grid cell scale, we looked along sub-national political boundary levels to determine how measles susceptibility clusters within countries. To account for the nesting of administrative levels, we used a multi-level modelling framework for $p_k$, the expected probability of being vaccinated at grid cell $k$ from the GAM (for a given age, though the choice of the value used has no effect) with random effects for the three largest sub-national administrative (Adm) levels in the country[73]. The model used is shown in equation (2)

$$\text{logit}(p_k) = (1 \,|\, \text{Adm 1}) + (1 \,|\, \text{Adm 2}) + (1 \,|\, \text{Adm 3}) \qquad (2)$$

Zambia and Zimbabwe have only two Adm levels, so the last term of equation (2) was omitted for these two countries. We decomposed overall variance by estimating the Adm level-specific intraclass correlation coefficients (ICCs), which represent the proportion of overall variance in each country that is explained by that sub-national political boundary level. 95% CIs on ICCs were obtained from the 2.5th and 97.5th quantiles of parametric bootstrap distributions with 1,000 iterations.

**Transnational clustering of susceptibility.** We also looked along national boundaries to determine how measles susceptibility clusters between countries. We again used a multi-level modelling framework for $p_k$, the expected probability of being vaccinated at grid cell $k$ from the GAM (for a given age, though again the choice of the value used has no effect) now across the entire region, with a random effect for country. The model used is shown in equation (3)

$$\text{logit}(p_k) = (1 \,|\, \text{country}) \qquad (3)$$

We estimated the ICC for country, which represents the proportion of overall variance in the entire region that is explained by national boundaries.

**Code availability.** All R code used in this analysis is available at: https://github.com/sakitakahashi/coldspots.

**Data availability.** All data used in this analysis is publicly available via ICF International[51–60], the WHO[63], the WorldPop project[66,67] and DIVA-GIS[68]. All relevant data are available from the authors upon request.

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

## Acknowledgements

This work is supported by the Bill & Melinda Gates Foundation (C.J.E.M., M.J.F., A.J.T., J.L.). A.J.T. is supported by funding from NIH/NIAID (U19AI089674), the Bill & Melinda Gates Foundation (OPP1106427, 1032350, OPP1134076), the Clinton Health Access Initiative, National Institutes of Health and a Wellcome Trust Sustaining Health Grant (106866/Z/15/Z).

## Author contributions

S.T., C.J.E.M., M.J.F., A.J.T. and J.L. conceived and designed the analysis. S.T. performed the data analysis. S.T., C.J.E.M. and J.L. wrote the first draft of the manuscript. S.T., C.J.E.M., M.J.F., A.J.T. and J.L. contributed to the writing and editing of the manuscript.

## Additional information

**Competing interests:** The authors declare no competing financial interests.

