## [Peer Review File · Nature Communications]

Reviewers' comments:

Reviewer #1 (Remarks to the Author):

Measles vaccination in the African Great Lakes region

The major claims

By using the GPS data available for some DHS surveys, the authors have produced detailed contour maps which highlight areas with large proportions, and numbers, of children unvaccinated against measles in 10 countries in Africa. They show that these 'coldspots' cross national and sub-national boundaries. On this basis they suggest a need for coordination between national and sub-national programmes.

Are the claims novel?

It is well known that control and elimination of measles require high levels of vaccine coverage, and that the coverage of vaccination programmes can be highly variable within countries (eg Cutts FT, Markowitz LE, 1994; Sartorius B Cohen C Chirwa T, 2013 - more details available if required). A study of variation in measles vaccine coverage between districts in Mozambique (Muloliwa, Artur Manuel et al 2013) came to similar conclusions about the need for coordination across administrative boundaries, but this did not produce smoothed contour maps and was for only one country. The authors refer to the Malaria Atlas project which has produced smoothed contour maps of malaria burden for Africa, and I have done an accompanying analysis of GPS-linked DHS data on vaccination coverage to overlay these, but I did not fit spatial smoothing models and the results have not yet been published.

I do think that this paper is well above the novelty threshold for publication.

Will the paper be of interest to others in the field?

Yes, but to specialists. I think a more specialist journal such as Nature Communications is the appropriate place within the Nature stable. However I don't know how widely this journal is read by people in the vaccine delivery field.

Will the paper influence thinking in the field?

I suspect it will, but I would be surprised if it led to major changes in policy on the ground. The results are hardly counter-intuitive. However it does provide very useful quantification and visualisation. I hope that this would provide focus and stimulate activity, and strengthen the hand of those already concerned.

Are the claims convincing?

In general the analysis is persuasive. I do have a few questions.

Line 49 (a small point). Critical community size is quoted for England & Wales as 300,000. Might this be smaller in Africa, with much higher proportions of young children?

The authors provide estimates of uncertainty for their coverage estimates. This must lead to uncertainty about where the boundaries of coldspots lie. The mapping of boundaries is very appealing but does not indicate this uncertainty, and I am not sure how it could, other than providing different maps based on, say upper and lower ranges of coverage estimates. There could be very substantial boundary shifts in low density areas.

Line 96: 'We estimated measles vaccination coverage at 24 months of age'. Does this mean for the group of children aged 24-59 months? If not, how was this calculated?

Line 98. The authors include survey data on vaccination during campaigns, as they should. However as I understand it, these vaccinations are seldom recorded on the vaccination card. The survey data on vaccinations in campaigns generally depend on mother's recall, and do not have dates of administration and hence child's age at administration. As far as I can make out the authors used data on known campaigns to date these doses, which they need to estimate age-specific coverage. This would work well where campaigns are organized as national vaccination days. Otherwise it may not be so accurate. More explanation and comment needed.

Line 111. "Subnational political boundaries are shown in light grey". These look like national boundaries to me.

Line 134 et seq. The implication seems to be that absolute number of unvaccinated children in an area is a good indicator of risk. However I would have thought that population density would also make a contribution. For areas of equal size, would the chances of child A mixing with child B not be lower in a large population than in a small one?

Line 167. Spell out GAM here (this is done later in line 312).

Line 170: 'only 54% of the variation in the probability ... is explained by the country [*my italics*]. I am surprised that it is this much!

Lines 252-254: 'in ways that might increase the absolute burden or degree of inequity .." This statement should be explained – or if this is too complicated, dropped.

Line 278. Excluding children aged < 6m is fine but the vaccination schedule target age for MCV1 is generally 9 months. Thus one would expect children not involved in campaigns and aged < 9 months to be unvaccinated. Would this not bias the coverage estimates downwards? Nonetheless the authors used 80% coverage as the critical level. However if the 80% is only applied to the larger group (6-59m) this bias will not make much difference.

I am guessing that the calculation of rates at 24m was done by restricting the analysis to children aged 24m or more. Is this right? (See my query for line 96 above.)

Line 327. What was the distribution of numbers of children covered across the 10x10km grid squares? Ie what % of grid squares contained small numbers of children?

P26: Table 3: description of DHS and SIA data. I would like to see two more columns in this table:

1: % of children in the survey with complete dates of birth. This can be quite low in some countries.

2: % of vaccinations ($v_i = 1$ in the modelling) based on vaccination card data rather than mother's report.

Supporting material

It would make life much easier for the reader if the legend for each figure was on the same page as the figure itself!

Other questions

In general this is a well-written paper on an important topic with practical implications. It is to the point without being over-compressed. I have asked for more explanation on one or two matters but this does not mean that the paper has to become longer.

Re the amount of methodological detail, I have indicated where I think we could do with a little more.

I am not an expert on spatial or hierarchical modeling, but I do have a general knowledge of these subjects and I have not noticed anything that suggests that the analysis might be unsound.

No ethical concerns.

Reviewer #2 (Remarks to the Author):

This paper is an interesting presentation of the levels of measles immunization coverage in several subsaharan countries in Africa. The argument is primarily descriptive, and its conclusions are not new. However, the results are presented in an interesting and useful way. The results are useful to people working in the field, and gather together and synthesize data that are usually not that easy to digest. The paper shows suboptimal measles immunization coverage in many areas, that has persisted for some time. It shows that measles elimination will not be achieved if the programme continues at current levels. The authors could strengthen arguments about secular trends. They define and discuss 'long-term coldspots', but these are defined in terms of cross-sections with ages between 24 and 60 months. Effective SIAs targeting all children ≤ 60 months should mop up these areas. My concern is the proportion of children who escape to older ages without immunity:

coverage is high enough to alter the measles attack risk in the short term, but not high enough in the coldspots to prevent outbreaks in adolescents and adults in 10 to 20 years time.

I think this kind of presentation of routine data can be very helpful in decision-making in immunization programmes. I particularly like the idea of combining coverage (as a fraction) with numbers of children at risk. It is important to focus programmes more on areas where gains can be made, and low coverage in remote areas with few children is less important epidemiologically than low coverage in densely populated areas.

I can't comment on the statistics, as I am not a mapping expert, but the data manipulation and analysis looks reasonable to me.

Some minor comments:

44 Measles is also transmissible in infected airspaces for some hours after an infective has left, so the definition of contact is not exactly direct transmission, though this has little impact on the arguments presented in the paper

54 We often use spatial mixing as a surrogate for the social mixing that is the real driver of transmission. Within the geographical areas there will be situations of higher, or lower, intensity of transmission; for example schools. The authors mention spatial heterogeneity, but there are other important heterogeneities of mixing. Socially defined coldspots may be important.

73 This appears to be a classic honeymoon period to me.

134 The interaction of secular changes and age is very important, as I have argued above. Children escaping to adolescence with no immunity (not from vaccine nor from wild infection) will become the focus of adolescent and adult measles outbreaks in the future.

157 The data show very clear country-level effects as well as province-level effects. Some countries in this series do markedly better. Because any improvement in immunization coverage will have to be driven by the immunization programme it is important to identify programmatic weaknesses. Geographic data may help here, but there is a plethora of management and social reasons why coverage may be low, some examples include vaccine stockouts, funding crises, or gender differences in uptake. It is important that the seductiveness of a geographical presentation doesn't hide other reasons.

Robert Hall

Reviewer #3 (Remarks to the Author):

The paper's major claims are to estimate measles vaccination coverage vs. age at subnational spatial scales across subnational boundaries. The paper defines regions

("coldspots") independent of subnational boundaries where vaccination fails to reach the WHO target of 80% coverage, estimates the total population of unvaccinated children under 5, and calls attention to regions of high population density but low vaccination coverage that are critical to measles elimination goals.

The central claim, that the relevant scales of heterogeneity in vaccination coverage are not always well-captured when administrative boundaries (particularly at country or province-level) are used as areal units for averaging, is of interest to global and country policymakers and to researchers in measles and other vaccine-preventable diseases. The results should influence thinking in the field with regards to planning and monitoring campaigns and tracking vaccination activities at subnational scales.

The results are novel. A search of the literature reveals no obviously similar studies; where MCV1 coverage is estimated, it is generally done so averaging over administrative boundaries at country/province/district level. The authors convincingly demonstrate the existence of vaccination coldspots with high population density whose structure is not captured well when averaged over high-level administrative boundaries.

I recommend the paper be accepted for publication, pending revisions.

Comments to the authors:

Major comments:

1. My experience with the DHS child surveys is that the child samples are self-weighted within clusters, but that the surveyed clusters have associated sample weights. I found no mention of whether the sample weights played a role in estimation of the GAM or associated uncertainties. If they were used, I think it should be mentioned; if not, it bears acknowledgement and an argument for why the results should still hold.

2. Do the authors plan to publish any code developed for this data analysis? I did not see an indication of this. The primary analysis techniques appear to be available in publicly available R packages, and the data are all publicly available as well. However, in the interest of easing the reproducibility of the results, it would be nice to see the publication of any code employed for data cleaning and preparation, and the configuration of the data analysis methods. A relevant reference:

"Papers in Nature journals should make computer code accessible where possible." *Nature* 514, 536 (30 October 2014) doi:10.1038/514536a

3. The literature contains other examples of spatial interpolation of DHS data (with or without other spatial covariates), and a brief (paragraph or two) review highlighting some of this literature and comparing the GAM method chosen here against the options chosen by other researchers would strengthen the methods section. A couple of examples found in a quick Google Scholar search:

"Gemperli A, Vounatsou P, Kleinschmidt I, Bagayoko M, Lengeler C, Smith T. Spatial Patterns of Infant Mortality in Mali: The Effect of Malaria Endemicity. *Am J Epidemiol* 2004; 159: 64–72."

"Soares Magalhães RJ, Clements ACA (2011) Mapping the Risk of Anaemia in Preschool-Age Children: The Contribution of Malnutrition, Malaria, and Helminth Infections in West Africa. *PLoS Med* 8(6): e1000438. doi:10.1371/journal.pmed.1000438"

"Gething, Peter, Andy Tatem, Tom Bird, and Clara R. Burgert-Brucker. 2015. *Creating Spatial Interpolation Surfaces with DHS Data* DHS Spatial Analysis Reports No. 11. Rockville, Maryland, USA: ICF International."

4. In Methods: Sub-national clustering of susceptibility. I'd like to see the equations specifying the multi-level model used to estimate the ICCs would strengthen this section, similar to Eq. 1 in the previous subsection specifying the GAM. Were the covariates used in the GAM included in this multi-level modeling, or does the model include only effects for country/province/district?

5. Lines 154-155: "Model residual maps do not suggest existence of unmodeled spatial autocorrelation (Supplementary Fig. 5)". Can this assertion be tested in a more formal way? I'd also suggest trying a different color scheme for these supplementary figures; I don't find the negative and positive residuals particularly distinguishable from each other.

Minor comments:

1. Line 31: grammatical error - "40 cases per million population cases".

2. The authors note that DHS cluster coordinates are randomly displaced by 2/5 km in urban/rural areas, and state that "For the purposes of this analysis we assume cluster locations are exact". I expect that the effects of this assumption are small, given that the scale of displacement is small relative to the scale of the multi-country analysis. However, if the scale of any "displacement effects" on the outcome maps have been addressed in previous literature, it would be worth a reference. If not, but the authors have demonstrated to themselves that the effects are small, it may be worth a figure or brief discussion in the supplement.

3. A clarification in the Methods on Estimating Vaccination Coverage. Line 325: "Model selection was assessed based on Akaike Information criterion (AIC) (Supplementary Table 1, Supplementary Table 2)." As I read these tables, the selected model for Burundi and Tanzania actually excludes subnational SIAs, so that DRC is the only country in which the selected model includes subnational SIA eligibility. I think a clarifying statement indicating this would be appropriate.

Reviewers' comments:

Reviewer #1 (Remarks to the Author):

Measles vaccination in the African Great Lakes region

The major claims

By using the GPS data available for some DHS surveys, the authors have produced detailed contour maps which highlight areas with large proportions, and numbers, of children unvaccinated against measles in 10 countries in Africa. They show that these 'coldspots' cross national and sub-national boundaries. On this basis they suggest a need for coordination between national and sub-national programmes.

Are the claims novel?

It is well known that control and elimination of measles require high levels of vaccine coverage, and that the coverage of vaccination programmes can be highly variable within countries (eg Cutts FT, Markowitz LE, 1994; Sartorius B Cohen C Chirwa T, 2013 - more details available if required). A study of variation in measles vaccine coverage between districts in Mozambique (Muloliwa, Artur Manuel et al 2013) came to similar conclusions about the need for coordination across administrative boundaries, but this did not produce smoothed contour maps and was for only one country. The authors refer to the Malaria Atlas project which has produced smoothed contour maps of malaria burden for Africa, and I have done an accompanying analysis of GPS-linked DHS data on vaccination coverage to overlay these, but I did not fit spatial smoothing models and the results have not yet been published.

I do think that this paper is well above the novelty threshold for publication.

Will the paper be of interest to others in the field?

Yes, but to specialists. I think a more specialist journal such as Nature Communications is the appropriate place within the Nature stable. However I don't know how widely this journal is read by people in the vaccine delivery field.

Will the paper influence thinking in the field?

I suspect it will, but I would be surprised if it led to major changes in policy on the ground. The results are hardly counter-intuitive. However it does provide very useful quantification and visualisation. I hope that this would provide focus and stimulate activity, and strengthen the hand of those already concerned.

Are the claims convincing?

In general the analysis is persuasive. I do have a few questions.

Line 49 (a small point). Critical community size is quoted for England & Wales as 300,000. Might this be smaller in Africa, with much higher proportions of young children?

The high birth rates in Africa and differences in population density (i.e., areas of the African Great Lakes region include some of the most densely populated places in the world), as well as other factors may impact the critical community size in predictable and unpredictable ways. For instance, a study by Ferrari et al (2008) *Nature* estimated the critical community size of measles in Niamey, Niger to be well over a million due to the strong seasonality in transmission. We still feel that the critical community size in England and Wales is a useful reference point, but now highlight the potential differences in the African context in this sentence, along with a reference to Ferrari et al (2008):

Even when the size of these unvaccinated clusters is below the critical community size required to maintain measles transmission (estimated to be around 300,000 individuals for measles in pre-vaccination England and Wales⁹, though differences in birth rate and population density may lead to differences in the African context¹⁰), they remain a concern: [...]

The authors provide estimates of uncertainty for their coverage estimates. This must lead to uncertainty about where the boundaries of coldspots lie. The mapping of boundaries is very appealing but does not indicate this uncertainty, and I am not sure how it could, other than providing different maps based on, say upper and lower ranges of coverage estimates. There could be very substantial boundary shifts in low density areas.

We agree that it is important to characterize uncertainty in coldspot boundaries. To do so, we have created Supplementary Fig. 9: this figure shows the confidence in our classification of each grid cell as a coldspot. It synthesizes our maps of the mean coverage (Fig. 1a and Supplementary Fig. 6), maps of coldspot boundaries (Fig. 1b and Supplementary Fig. 7), and maps of the estimated standard error of the mean coverage (Supplementary Fig. 8) for different ages.

Specifically, the values of the maps in Supplementary Fig. 9 represent the proportion of the distribution of mean coverage at various ages that is below 80% (i.e., estimated to be a coldspot), assuming that the distribution of mean coverage at each grid cell is normal and using the estimates and the standard errors of model predictions on the logit scale. On each map in Supplementary Fig. 9, we have also overlaid the outline of the estimated coldspot boundaries in

black. We have also added a passing reference to Supplementary Fig. 9 in the ‘Coldspots of vaccination’ sub-section of the Results section of the main text. We believe that this new figure is useful for visualizing coldspots beyond the binary classification, and thank the reviewer for this suggestion.

Line 96: ‘We estimated measles vaccination coverage at 24 months of age’. Does this mean for the group of children aged 24-59 months? If not, how was this calculated?

Since we are estimating a smoothed monthly effect of age (Supplementary Fig. 4), coverage at 24 months refers to the probability of being vaccinated at that exact month of age. To clarify this point, we have made the following changes to (1) the ‘Vaccination coverage’ sub-section of the Results section, and (2) the ‘Estimating vaccination coverage’ sub-section of the Methods section:

- (1) *We estimated measles vaccination coverage at each month of age across the region using generalized additive models (GAMs)²³ (see Methods) from DHS surveys conducted between 2009-2014. Estimated vaccination coverage is the result of both routine immunization programs and national SIA campaigns for which children are eligible, given their age. Taking vaccination coverage at 24 months of age as a representative scenario, results indicate large contiguous areas of low vaccination coverage across the region (Fig. 1a).*
- (2) *Monthly survey age was also included as a smoothed covariate, and showed a broadly increasing relationship with vaccination coverage (Supplementary Fig. 4, Supplementary Table 3).*

We note that there was a coding error in the original submission which erroneously included children under 6 months of age in the analysis. This had no qualitative effect on the results, since the monthly effect of age before 6 months was negligible. We apologize for the error, and have updated the figures and table values throughout the manuscript and supplement to reflect this correction.

Line 98. The authors include survey data on vaccination during campaigns, as they should. However as I understand it, these vaccinations are seldom recorded on the vaccination card. The survey data on vaccinations in campaigns generally depend on mother’s recall, and do not have dates of administration and hence child’s age at administration. As far as I can make out the authors used data on known campaigns to date these doses, which they need to estimate age-specific coverage. This would work well where campaigns are organized as national vaccination days. Otherwise it may not be so accurate. More explanation and comment needed.

We agree that the lack of vaccine date in maternal recall is a limitation of the data. In our model, we do not use any information on a child's vaccine date: our outcome variable is whether the child had ever received a measles vaccine (based on vaccination card or report of parent/guardian), and the covariates from the DHS survey are the GPS location and the child's age at the time of survey (rounded up into 1-month classes due to uncertainty in the data). Based on information on the dates that sub-national SIA campaigns occurred, we then infer the eligibility of each child for each campaign based on their age, and use that as an additional covariate in the GAM model. We further clarify this point in the 'Data' sub-section of the Methods section:

We do not use the information on a child's vaccine date in this analysis.

We note that there was a data error in the original submission, which included erroneous SIA campaign information for DRC in 2009 and 2011. We obtained updated information from Scobie et al (2015) *Pan Afr Med J* (see Table 1 and Supplementary Table 4), which we have confirmed to be more accurate than our previous source. We have included a citation to this new reference. We also note that previously we had that the sub-national SIA campaign in Tanzania in 2008 targeted all of the country except for Zanzibar, but we have now also correctly removed individuals in Pemba Island from eligibility as well. We apologize for the errors, and have updated the figures and table values throughout the manuscript and supplement to reflect this correction.

Line 111. "Subnational political boundaries are shown in light grey". These look like national boundaries to me.

In the original Fig. 1b (as well as Fig. 3a and Fig. 3b), the light grey was too difficult to see. We have changed the coloring to make this distinction more obvious in our figures throughout the manuscript and supplement.

Line 134 et seq. The implication seems to be that absolute number of unvaccinated children in an area is a good indicator of risk. However I would have thought that population density would also make a contribution. For areas of equal size, would the chances of child A mixing with child B not be lower in a large population than in a small one?

This is, to some extent, our point. As the reviewer notes, population density is one factor that influences measles risk. Because we are looking at fixed grid cell sizes, the reported results in Fig. 3b are a measure of the density of unvaccinated children, and so attempt to capture the combined effect of low vaccine coverage and population density (whereas figures with coldspots based solely on the vaccinated proportion – as in Fig. 3a – only capture the impact of

those rates). In addition, absolute numbers of unvaccinated children – as in Fig. 2 – may have a policy relevance independent of risk, as they highlight areas where vaccination efforts might have the most “bang for the buck”.

We have modified the paper in (1) the ‘Coldspots of vaccination and population density’ subsection of the Results section and (2) added a paragraph to the Discussion section to highlight these issues more explicitly:

(1) *However, Fig. 3a captures only the estimated levels of vaccination coverage and does not account for population density, which is another factor that influences measles risk. Therefore, to obtain a complete picture of what Fig. 3a indicates in the context of population size, we further partitioned the long-term coldspots into ‘low-density’ and ‘high-density’ areas: only coldspots with at least 500 children under 60 months of age per grid cell are shown in Fig. 3b.*

(2) *In addition to proportions unvaccinated by age (which directly translate to coldspots), population density is also an important factor that influences measles risk, since more densely populated areas are more likely to sustain epidemics than less densely populated areas⁹. Areas with both low vaccination coverage across ages and a high density of children are likely to have a large pool of susceptible individuals, and are thus at greatest risk for an outbreak. Furthermore, the absolute numbers of unvaccinated children (as illustrated in Fig. 2) may have a policy relevance independent of risk, as they highlight areas where targeting vaccination efforts might be most cost-effective²⁸.*

Line 167. Spell out GAM here (this is done later in line 312).

Corrected, thanks.

Line 170: ‘only 54% of the variation in the probability ... is explained by the country [my italics]. I am surprised that it is this much!

Good point, we have removed the ‘only’. We have also added confidence intervals on the ICC, obtained from the quantiles of a parametric bootstrap distribution (also added to Table 3).

Furthermore, we note that the corrections that we have made from above (i.e., removing children under 6 months of age from the analysis and the updated information on SIA campaigns) led to a small change in this estimate from 54% to 59% (95% CI: 28%-73%).

Lines 252-254: ‘in ways that might increase the absolute burden or degree of inequity ..’ This statement should be explained – or if this is too complicated, dropped.

We have removed the statement on ‘absolute burden’ or ‘degree of inequity in the burden’ of CRS, since this is beyond the scope of the paper. We have simplified this paragraph as follows:

As previous work has shown, inequities in vaccination can lead to an increased risk of rubella infection in pregnant women and thus a higher burden of CRS⁴⁸. Since rubella-containing vaccine is usually given in conjunction with measles-containing vaccine, the spatial heterogeneity in measles vaccination coverage documented here could affect the dynamics of rubella in ways that might increase the burden of CRS in this region^{49,50}.

Line 278. Excluding children aged < 6m is fine but the vaccination schedule target age for MCV1 is generally 9 months. Thus one would expect children not involved in campaigns and aged < 9 months to be unvaccinated. Would this not bias the coverage estimates downwards? Nonetheless the authors used 80% coverage as the critical level. However if the 80% is only applied to the larger group (6-59m) this bias will not make much difference.

We chose to include children 6 months of age and older, because some SIA campaigns have their lower age target at 6 months (Supplementary Table 4 and Supplementary Table 11). Since we estimate coverage as the expected probability of vaccination at a particular age, the inclusion of these youngest children (e.g., those between 6-9 months of age) in our overall estimation procedure does not bias these results. For changes, please see response to comments above. We have also further clarified this point by adding the following to the beginning of the ‘Coldspots of vaccination’ sub-section of the Results section:

To define key areas of low coverage, we first defined a coldspot of vaccination as a grid cell that has below 80% estimated mean measles vaccination coverage at a given age (see Methods). We then mapped the coldspots of measles vaccination at 24 months of age, indicating areas from Fig. 1a where mean coverage is estimated to be below 80%, in grey (Fig. 1b).

I am guessing that the calculation of rates at 24m was done by restricting the analysis to children aged 24m or more. Is this right? (See my query for line 96 above.)

Please see response to comments above.

Line 327. What was the distribution of numbers of children covered across the 10x10km grid squares? ie what % of grid squares contained small numbers of children?

We have added a Supplementary Table 10, which shows the total number of 10 km x 10 km grid cells by country, and the number of grid cells with at least 100, 500, and 1,000 children under

60 months of age. The latter three columns correspond to the threshold population sizes for Supplementary Fig. 12a, Fig. 3b, and Supplementary Fig. 12b, respectively. We have added a reference to this new table in the 'Coldspots of vaccination and population density' sub-section of the Results section as follows:

The percentage of total grid cells with at least 500 children ranged from 7% in Zambia to 87% in Burundi (Supplementary Table 10).

P26: Table 3: description of DHS and SIA data. I would like to see two more columns in this table:

- 1: % of children in the survey with complete dates of birth. This can be quite low in some countries.
- 2: % of vaccinations ($v_i = 1$ in the modelling) based on vaccination card data rather than mother's report.

We have re-ordered the tables in the main text such that this table is now Table 1. To address the comments:

1: Our analysis includes only those children for whom we had a known month and year of birth. Upon re-checking the raw DHS survey data to prepare this response, we found that the countries included in this analysis do not have any missing data for the month and year of birth. We now include a Supplementary Table 9, which details the DHS data processing steps that we took in this analysis to obtain the final data set for each country. The last column of this table matches with the number of children listed in Table 1.

2: Added as suggested.

Supporting material

It would make life much easier for the reader if the legend for each figure was on the same page as the figure itself!

We have restructured the supplement as suggested.

Other questions

In general this is a well-written paper on an important topic with practical implications. It is to the point without being over-compressed. I have asked for more explanation on one or two matters but this does not mean that the paper has to become longer.

Re the amount of methodological detail, I have indicated where I think we could do with a little more.

I am not an expert on spatial or hierarchical modeling, but I do have a general knowledge of these subjects and I have not noticed anything that suggests that the analysis might be unsound.

No ethical concerns.

Thank you very much for the thoughtful comments.

Reviewer #2 (Remarks to the Author):

This paper is an interesting presentation of the levels of measles immunization coverage in several subsaharan countries in Africa. The argument is primarily descriptive, and its conclusions are not new. However, the results are presented in an interesting and useful way. The results are useful to people working in the field, and gather together and synthesize data that are usually not that easy to digest. The paper shows suboptimal measles immunization coverage in many areas, that has persisted for some time. It shows that measles elimination will not be achieved if the programme continues at current levels. The authors could strengthen arguments about secular trends. They define and discuss 'long-term coldspots', but these are defined in terms of cross-sections with ages between 24 and 60 months. Effective SIAs targeting all children ≤ 60 months should mop up these areas. My concern is the proportion of children who escape to older ages without immunity: coverage is high enough to alter the measles attack risk in the short term, but not high enough in the coldspots to prevent outbreaks in adolescents and adults in 10 to 20 years time.

We have adjusted the manuscript to give a more in-depth treatment of these issues, as detailed in our responses to specific reviewer comments below.

I think this kind of presentation of routine data can be very helpful in decision-making in immunization programmes. I particularly like the idea of combining coverage (as a fraction) with numbers of children at risk. It is important to focus programmes more on areas where gains can be made, and low coverage in remote areas with few children is less important epidemiologically than low coverage in densely populated areas.

I can't comment on the statistics, as I am not a mapping expert, but the data manipulation and analysis looks reasonable to me.

Some minor comments:

44 Measles is also transmissible in infected airspaces for some hours after an infective has left, so the definition of contact is not exactly direct transmission, though this has little impact on the arguments presented in the paper

We have modified the text to better capture this nuance:

Measles is a virus transmitted mainly through direct contact (e.g., coughing and sneezing) and also by small-particle aerosols in an airspace where an infected person has coughed or sneezed in the previous few hours⁷. Infected individuals must enter into one of these forms of "contact" with susceptible individuals during the approximately two weeks that they are infectious in order for a chain of transmission to persist⁸.

54 We often use spatial mixing as a surrogate for the social mixing that is the real driver of transmission. Within the geographical areas there will be situations of higher, or lower, intensity of transmission; for example schools. The authors mention spatial heterogeneity, but there are other important heterogeneities of mixing. Socially defined coldspots may be important.

This is true: our group has also looked at effects of school-term on the seasonality of measles and other infections, and we agree that it is important for studying transmission dynamics. A detailed examination of this is beyond the scope of this paper and the capabilities of our data. However, we now explicitly mention this point in the Discussion section:

An additional limitation here is that we do not include aspects such as the effect of school-term on seasonality³² or the impact of schools as hotspots of transmission³³, both of which are sources of non-geographic heterogeneities of mixing that are important to understanding the dynamics of observed measles incidence.

73 This appear to be a classic honeymoon period to me.

It looks as if there might be multiple causes of these resurgences, including classic honeymoon period effects and the mentioned weakness in vaccination programs. We have now mentioned honeymoon period effects as a possible synergistic mechanism:

Such outbreaks have been attributed to weak routine vaccination systems and delayed or low-quality SIA campaigns²⁰, along with “honeymoon period” effects as a possible synergistic mechanism²¹. A lack of transnational coordination in the timing of SIAs may also contribute [...]

134 The interaction of secular changes and age is very important, as I have argued above. Children escaping to adolescence with no immunity (not from vaccine nor from wild infection) will become the focus of adolescent and adult measles outbreaks in the future.

We have emphasized this important point in the Discussion section as follows:

Age cohorts missed by routine vaccination in each year will allow continued circulation of the virus, unless they are removed from the pool of susceptibles by natural infection or a broader age range campaign like an SIA. Furthermore, insufficient vaccination coverage may increase the average age of infection such that children with no immunity (neither from vaccination nor from wild infection) will become the focus of adolescent and adult measles outbreaks in the future²⁷.

157 The data show very clear country-level effects as well as province-level effects. Some countries in this series do markedly better. Because any improvement in immunization coverage will have to be driven by the immunization programme it is important to identify programmatic weaknesses. Geographic data may help here, but there is a plethora of management and social reasons why coverage may be low, some examples include vaccine stockouts, funding crises, or gender differences in uptake. It is important that the seductiveness of a geographical presentation doesn't hide other reasons.

Very good point: we do not currently have the data to approach this question, but it is important. In fact, we are currently starting work aiming to address these issues and to also bring in other covariates (e.g., remoteness and accessibility) that may get at this question. We now emphasize this more in the Discussion section as follows:

Methodological refinements in a model-based geostatistical framework (e.g., the Malaria Atlas Project³⁴ and the WorldPop project³⁵) and the inclusion of spatial metrics such as urbanicity, remoteness, or accessibility³⁶ will strengthen model fit (Supplementary Table 1, Supplementary Table 2) and capture smaller-scale spatial variability that is missed by this current approach. [...] Additionally, it will be important to identify and quantify the effects of logistical issues such as vaccine stock outs³⁷ or socioeconomic factors³⁸ that may also affect the performance of immunization programs.

Robert Hall

Thank you very much for the thoughtful comments.

Reviewer #3 (Remarks to the Author):

The paper's major claims are to estimate measles vaccination coverage vs. age at subnational spatial scales across subnational boundaries. The paper defines regions ("coldspots") independent of subnational boundaries where vaccination fails to reach the WHO target of 80% coverage, estimates the total population of unvaccinated children under 5, and calls attention to regions of high population density but low vaccination coverage that are critical to measles elimination goals.

The central claim, that the relevant scales of heterogeneity in vaccination coverage are not always well-captured when administrative boundaries (particularly at country or province-level) are used as areal units for averaging, is of interest to global and country policymakers and to researchers in measles and other vaccine-preventable diseases. The results should influence thinking in the field with regards to planning and monitoring campaigns and tracking vaccination activities at subnational scales.

The results are novel. A search of the literature reveals no obviously similar studies; where MCV1 coverage is estimated, it is generally done so averaging over administrative boundaries at country/province/district level. The authors convincingly demonstrate the existence of vaccination coldspots with high population density whose structure is not captured well when averaged over high-level administrative boundaries.

I recommend the paper be accepted for publication, pending revisions.

Comments to the authors:

Major comments:

1. My experience with the DHS child surveys is that the child samples are self-weighted within clusters, but that the surveyed clusters have associated sample weights. I found no mention of whether the sample weights played a role in estimation of the GAM or associated uncertainties. If they were used, I think it should be mentioned; if not, it bears acknowledgement and an argument for why the results should still hold.

We did not use the DHS sample weights since it is not clear they are appropriate in this type of spatial analysis, as we are not calculating aggregate or national measures. However, we now include the DHS sample weights as a sensitivity analysis in Supplementary Fig. 17, showing that they have no qualitative impact on the results. We have clarified this in the 'Estimating vaccination coverage' sub-section of the Methods section as follows:

We do not use the DHS sample weights here, as we do not calculate any aggregate or national measures (robustness to sample weights shown in Supplementary Fig. 17).

2. Do the authors plan to publish any code developed for this data analysis? I did not see an indication of this. The primary analysis techniques appear to be available in publicly available R

packages, and the data are all publicly available as well. However, in the interest of easing the reproducibility of the results, it would be nice to see the publication of any code employed for data cleaning and preparation, and the configuration of the data analysis methods. A relevant reference:

"Papers in Nature journals should make computer code accessible where possible." Nature 514, 536 (30 October 2014) doi:10.1038/514536a

Yes, thank you for bringing up this point. We have made all of the R source code for this analysis available on GitHub, with a link to the repository in the 'Code Availability' sub-section of the Methods section.

3. The literature contains other examples of spatial interpolation of DHS data (with or without other spatial covariates), and a brief (paragraph or two) review highlighting some of this literature and comparing the GAM method chosen here against the options chosen by other researchers would strengthen the methods section. A couple of examples found in a quick Google Scholar search:

"Gemperli A, Vounatsou P, Kleinschmidt I, Bagayoko M, Lengeler C, Smith T. Spatial Patterns of Infant Mortality in Mali: The Effect of Malaria Endemicity. Am J Epidemiol 2004; 159: 64–72."

"Soares Magalhães RJ, Clements ACA (2011) Mapping the Risk of Anaemia in Preschool-Age Children: The Contribution of Malnutrition, Malaria, and Helminth Infections in West Africa. PLoS Med 8(6): e1000438. doi:10.1371/journal.pmed.1000438"

"Gething, Peter, Andy Tatem, Tom Bird, and Clara R. Burgert-Brucker. 2015. Creating Spatial Interpolation Surfaces with DHS Data DHS Spatial Analysis Reports No. 11. Rockville, Maryland, USA: ICF International."

While the reviewer's point is taken that there has been significant work making smooth surfaces from DHS data in the past, we feel that a discussion of this in detail would require its own paper. However, first we have now increased the number of citations of GAM-based methodology analogous to ours in the Methods section, including a paper referenced above (Gething et al (2015) *DHS Spatial Analysis Reports*) that make comparisons between the GAM and other models for spatial interpolation using DHS data.

Second, we also now in the Discussion section mention other interpolation papers, to highlight both the public health utility of such methods applied to survey data and the multitude of spatial smoothing approaches available:

Methodological refinements in a model-based geostatistical framework (e.g., the Malaria Atlas Project³⁴ and the WorldPop project³⁵) and the inclusion of spatial metrics such as urbanicity, remoteness, or accessibility³⁶ will strengthen model fit

(Supplementary Table 1, Supplementary Table 2) and capture smaller-scale spatial variability that is missed by this current approach. [...] The increasing abundance of methods development and applications of spatial interpolation for DHS and similar data³⁹⁻⁴⁴ broadly highlight the utility of maps as a tool for public health planning and advocacy. For example, the DHS Program has recently created a Spatial Data Repository^{45,46} with interpolated surfaces of health indicators, including measles vaccination coverage among children 12-23 months of age, in a Bayesian model-based geostatistical framework with spatial covariates. While we use the same data source in this analysis to model a similar outcome, our approach here differs in that we fit an age-varying probability of vaccination.

4. In Methods: Sub-national clustering of susceptibility. I'd like to see the equations specifying the multi-level model used to estimate the ICCs would strengthen this section, similar to Eq. 1 in the previous subsection specifying the GAM. Were the covariates used in the GAM included in this multi-level modeling, or does the model include only effects for country/province/district?

We have added the equation here as suggested as Eq. 2 (and Eq. 3 for the added 'Transnational clustering of susceptibility' sub-section). As now shown in Eq. 2, the model only includes effects for the three largest sub-national administrative levels in each country.

5. Lines 154-155: "Model residual maps do not suggest existence of unmodeled spatial autocorrelation (Supplementary Fig. 5)". Can this assertion be tested in a more formal way? I'd also suggest trying a different color scheme for these supplementary figures; I don't find the negative and positive residuals particularly distinguishable from each other.

Indeed: we have re-done the analysis of residual spatial autocorrelation in an updated Supplementary Fig. 5, where we now show the variogram of standardized residuals for each country in a more rigorous test for spatial autocorrelation. The variogram of residuals shows minimal autocorrelation, which indicates that most of the spatial structure was accounted for in the GAM model.

Minor comments:

1. Line 31: grammatical error - "40 cases per million population cases".

Corrected, thanks.

2. The authors note that DHS cluster coordinates are randomly displaced by 2/5 km in urban/rural areas, and state that "For the purposes of this analysis we assume cluster locations are exact". I expect that the effects of this assumption are small, given that the scale of displacement is small relative to the scale of the multi-country analysis. However, if the scale of any

“displacement effects” on the outcome maps have been addressed in previous literature, it would be worth a reference. If not, but the authors have demonstrated to themselves that the effects are small, it may be worth a figure or brief discussion in the supplement.

We now include a sensitivity analysis as Supplementary Fig. 16 in which we add random displacement (up to 2 km in urban areas and up to 5 km in rural areas) to the cluster locations and re-fit the models. We find that the effect of DHS cluster displacement is negligibly small. We have updated the sentence above as follows:

For the purposes of this analysis we assume cluster locations are exact. In sensitivity analysis we find the effects of this random displacement to be negligibly small (Supplementary Fig. 16).

3. A clarification in the Methods on Estimating Vaccination Coverage. Line 325: “Model selection was assessed based on Akaike Information criterion (AIC) (Supplementary Table 1, Supplementary Table 2).” As I read these tables, the selected model for Burundi and Tanzania actually excludes subnational SIAs, so that DRC is the only country in which the selected model includes subnational SIA eligibility. I think a clarifying statement indicating this would be appropriate.

The ‘Full model’ from Supplementary Table 1 and 2, which includes the sub-national SIA covariate(s), is what we used in the main analysis. We have clarified this in the Methods section as follows, and have added an explanation in the legend for Supplementary Table 1 as well:

Equation (1) reflects the primary model used in this analysis, and the Akaike information criterion (AIC) values of sub-models is presented in Supplementary Table 1 and Supplementary Table 2 (also see Supplementary Fig. 14).

The ‘Full model’ was chosen a priori based on the assumed epidemiological impact of sub-national SIA campaigns, but we have also added a sensitivity analysis removing the sub-national SIA covariate in Burundi and Tanzania (where the AIC was marginally lower in this model) and re-fitting the models as Supplementary Fig. 14, where we see no qualitative differences in the results.

Thank you very much for the thoughtful comments.

REVIEWERS' COMMENTS:

[Reviewer #1 had no remarks to the author.]

Reviewer #2 (Remarks to the Author):

I think the authors have done a good job in responding to the reviewers' comments, and despite the complexities introduced by our comments and suggestions the paper reads well and coherently. I think that the arguments are sound and that the paper is of sufficient merit to be published.

Robert Hall

Reviewer #3 (Remarks to the Author):

I thank the authors for satisfactorily addressing each of my comments from the original review, and recommend the paper for publication.